# Postpartum Assessment of the Correlation between Serum Hormone Levels of Estradiol, Progesterone, Prolactin and ß-HCG and Blood Pressure Measurements in Pre-Eclampsia Patients

**DOI:** 10.3390/diagnostics12071700

**Published:** 2022-07-12

**Authors:** Mariz Kasoha, Zoltan Takacs, Jacob Dumé, Sebastian Findeklee, Christoph Gerlinger, Romina-Marina Sima, Liana Ples, Erich-Franz Solomayer, Bashar Haj Hamoud

**Affiliations:** 1Department of Gynecology, Obstetrics and Reproductive Medicine, University Medical School of Saarland, 66421 Homburg, Germany; zoltan.takacs@uks.eu (Z.T.); jacob.dume@uks.eu (J.D.); sebastian.findeklee@uks.eu (S.F.); c.gerlinger@gmx.de (C.G.); erich.solomayer@uks.eu (E.-F.S.); bashar.hajhamoud@uks.eu (B.H.H.); 2Department of Obstetrics and Gynecology, ‘Carol Davila’ University of Medicine and Pharmacy, 020021 Bucharest, Romania; romina.sima@yahoo.es (R.-M.S.); liana.ples@umfcd.ro (L.P.)

**Keywords:** preeclampsia, hypertension, hormones, postpartum, serum

## Abstract

**Background:** Preeclampsia is a pregnancy-related hypertensive disease. Aberrant hormone levels have been implicated in blood pressure disorders. This study investigated the association of postpartum maternal serum hormone levels of estradiol, progesterone, prolactin, and ß-HCG with poorer PE-related complications including arterial hypertension. **Methods:** Thirty patient women with preeclampsia, and twenty women with uncomplicated pregnancy were included in this study. Serum levels of estradiol, progesterone, prolactin, and ß-HCG were determined immediately after delivery, and on the first and third postpartum days by means of ECLIA. **Results:** Compared with normal pregnancy cases, preeclampsia cases had higher serum levels of ß-HCG levels on Day-0 (319%), of progesterone on Day-0 (207%) and Day-1 (178%), and of estradiol on Day-1 (187%) and Day-3 (185%). Increased prolactin levels were positively associated with disease severity and estradiol and progesterone levels were decreased in poorer preeclampsia features including disease onset and IUGR diagnosis. No significant correlation between different hormone levels and blood pressure measurements was reported. **Conclusions:** This study is the first that detected postpartum maternal serum hormone levels and their correlation with blood pressure measurements in preeclampsia. We believe that the persistent arterial hypertension in the puerperium in preeclampsia as well as poorer disease specifications are most likely not of hormonal origin. Larger, well-defined prospective studies are recommended.

## 1. Introduction

Preeclampsia (PE) is a pregnancy-related hypertensive disease that is characterised by elevated arterial blood pressure (BP) (≥140/90 mmHg) associated with proteinuria (≥300 mg/dL in 24-h urine) and/or other organ failure, beginning after the 20th week of gestation [1]. Approximately 5% of pregnant women develop PE, which in turn causes almost 40% of early-term deliveries and is a major cause of maternal and fetal mortality in low-income countries [2]. Despite the amount of effort put into understanding PE, its pathophysiology and pathogenesis are still not fully understood. One of the more plausible and widely accepted theories is that trophoblast invasion is hampered early in pregnancy, resulting in impaired spiral artery remodelling in the placenta and restricted blood flow, resulting in a reduced nutrient supply to the foetus and signs of foetal growth restriction (FGR) [3]. Furthermore, it has been demonstrated that incomplete spiral artery remodelling in the uterus contributes to placental ischemia and, as a result, the release of antiangiogenic factors into the maternal circulation, resulting in endothelial damage [4]. PE can manifest in multiple organs, but it always results in arterial hypertension, which can last into the puerperium and increase the risk of future cardiovascular disease [5].

A hormonal genesis of PE-induced hypertension has been studied and obtained data is ambiguous. Female sex hormones including estradiol and progesterone are produced in the ovaries and in the placenta during pregnancy. Their blood levels increase steadily during pregnancy and continue to be high until birth. Estradiol and progesterone are crucial for a successful pregnancy via promoting and regulating different physiologic changes in the mother such as metabolic, cardiovascular, stimulating of mammary gland development, maternal tolerance of the feotus, and initiation of labour [6]. Maliqueo and colleagues showed that estradiol and progesterone act as regulators of uterine blood flow and the production of angiogenic factors in the placental tissue [7]. Therefore, abnormalities in these factors have been suggested as associated with different pregnancy complications including PE. Studies showed that decreased maternal serum estradiol levels appear to be associated with the development of PE [8,9]. In addition, exogenous estradiol supplementation had a BP-lowering effect in PE patients [10,11]. Furthermore, altered levels of progesterone were found to be involved in dysregulated trophoblastic proliferation and aberrant development of placental vasculature in PE patients [12,13].

Prolactin is a polypeptide hormone that has a significant role in the physiology of the breast and its level is imperative for normal lactation capabilities, with significant effects on the menstrual cycle also [14]. It is normally synthesized and secreted from the lactotrophs in the anterior pituitary gland. During early pregnancy, decidua and placenta secrete different types of prolactin-like molecules that can bind to the prolactin receptor, which in turn are involved in regulating pituitary prolactin secretion. This function is controlled by the local decidual prolactin-releasing factors and progesterone levels [15,16]. Apart from its role in maintaining pregnancy and lactation, prolactin has been implicated in immune system regulation, osmoregulation, and angiogenesis [17]. In normal pregnancy, the rise of maternal prolactin reaches 5–10 fold of its level in a non-normal state during the third trimester, and dysregulated maternal prolactin levels have been linked to pregnancy complications such as gestational diabetes and hypertension [18,19,20].

Beta-Human chorionic gonadotropin (β-HCG), better known as “pregnancy hormone”, is a glycoprotein produced by trophoblast cells. The main function of this hormone during pregnancy is to stimulate the production of progesterone and promote uterine and umbilical cord growth as well as placental development [21]. The association between aberrant β-HCG levels in the first trimester and the development of PE has been reported by several studies and β-HCG has been considered a serum marker for PE screening at 8–14 weeks of gestation [22,23,24]. 

To date, the only treatment for PE is delivery of the placenta, making this disease one of the leading causes of preterm birth. Therefore, the clinical use of several biochemical markers of placental dysfunction in combination with other relevant ultrasound and clinical manifestations provides the potential to improve pregnancy outcomes by allowing early detection and appropriate decision-making for referral of pregnant women at higher risk to proper care centres [25]. Effective screening and intervention strategies for preterm PE have been seriously considered. However, little is known about the value of maternal parameters in predicting PE complications postpartum [26]. The objective of this study is to investigate the correlation between maternal serum hormone levels of estradiol, progesterone, prolactin, and ß-HCG and their influence on the systolic and diastolic BP measurements immediately after delivery, on the first and the third postpartum days.

## 2. Materials and Methods

### 2.1. Study Cases and Performed Measurements

The current study included fifty women. Thirty women with PE as the patient group and twenty women with uncomplicated pregnancy as the control group. PE cases were obtained, respectively, from our blood bank. Control cases were prospectively enrolled in the study during their delivery in our department. 

All study cases were older than 18 years and had a gestational age between 23 + 5 and 41 + 6 weeks of gestation. Cases with chronic hypertension, chronic renal or lung disease, inflammatory or infectious disease, nicotine abuse, and gestations complicated by chromosomal or fetal anomalies were excluded.

Diagnosis of PE, determination of PE severity and onset, diagnosis of HELLP syndrome, and intrauterine growth retardation (IUGR) were carried out in accordance with the German national guideline for hypertensive illnesses during pregnancy, as described in our previous study [27]. PE was diagnosed using two criteria: blood pressure of 140/90 mmHg and proteinuria of more than 300 mg/24 h after the 20th week of pregnancy. At least one of the following parameters was required for the diagnosis of severe PE: progressive renal insufficiency (creatinine 0.9 mg/dL), blood pressure of 160/110 mmHg, thrombocytopenia, impaired liver function (elevated transaminases, upper abdominal pain), central nervous system dysfunction abnormalities (severe headache, blurred vision), pulmonary oedema, or IUGR. The presence of pathological Doppler of the umbilical and/or uterine artery and foetal estimated weight less than the 10th percentile indicated the diagnosis of IUGR. Late-onset PE was defined as PE patients with a gestational age ≥ 34 weeks. In addition, increased liver enzymes (transaminases > 35 U/L), decreased platelet count (thrombocytes < 100,000/µL), and lactate dehydrogenase [LDH] 2 times bigger than the upper level of the normal range [0–262 U/L]) or hemolysis (haptoglobin ≤ 25 mg/dL) indicated a diagnosis of HELLP syndrome.

Aseptic venous blood samples from control subjects were collected using serum gel monovette between 8 a.m. and 10 a.m. on the day of delivery immediately after giving birth and on the first and third day postpartum. Samples were centrifuged and supernatants were transferred in aliquots into Eppendorf tubes that were stored at −80° until they could be analysed. All clinical data were collected by reviewing electronic patients’ medical records. 

Serum levels of estradiol, progesterone, prolactin, and ß-HCG were tested in the Central Laboratory of our University hospital by means of electrochemiluminescence immunoassay (ECLIA) from Roche^®^ according to the manufacturer’s instructions. The technical variability of the hormonal assays ranged from 1.68–2.30% for ß-HCG, 2.07–2.55% for prolactin, 3.12–3.26% for progesterone, and 1.61–1.83% for estradiol. In addition, BP readings were taken regularly during blood sampling days using the BP monitor from the company Erka^®^. Measurements were performed in the early morning, at noon, in the afternoon, and in the evening and were recorded in the patient registry. The highest BP reading was used in the study. 

### 2.2. Statistical Analysis

Following the intent-to-treat principle, all patients that provided data at baseline were included in the analyses. As appropriate for explorative analyses comparison-wise two-sided significance level α of 5% was used. The statistical analyses were performed using RStudio version 1.2.5042 and R version 4.0.3 running under Ubuntu 20.04 LTS. The R packages gtsummary, and tidyverse were used.

## 3. Results

### 3.1. Clinical and Laboratory Parameters of Study Cases

Briefly, control and patient women were age-matched [Mean ± STD (years): 31 ± 7 vs. 32 ± 5, respectively, *p* > 0.05]. PE cases had higher BMI values [Mean ± STD: 36.9 ± 8.7 vs. 32.3 ± 5.7, respectively, *p* = 0.030] as well as higher creatinine, ALT, and AST concentrations. Furthermore, gestation age at delivery and number of gravida and para were significantly lower in PE cases compared with control cases. The existence of one or more pathological signs such as headache, severe edema, vaginal bleeding, epigastric pain, and vision changes was observed in 20% of PE cases (6/30) but not in control cases; however, this difference was not statistically significant. All parameters are displayed in Table 1.

### 3.2. Postpartum Serum Hormone Levels and BP Measurements in Normal and PE Cases

Serum hormone levels of estradiol, progesterone, prolactin, and ß-HCG were detected on the day of delivery immediately after giving birth (Day-0) and on the first day (Day-1) and third day (Day-3) postpartum. Blood samples of PE cases were available from all 30 subjects on Day-0, from 29 subjects on Day-1 and from 26 subjects on Day-3. In addition, BP measurements on Day-0, Day-1, and Day-3 were obtained from medical records of patients of 29, 28, and 26 subjects, respectively. On the other hand, all hormones and BP measurements of the control cases were performed for all 20 subjects on all study days.

As presented in Table 2, systolic and diastolic BP measurements were significantly higher in PE cases compared with control cases during all study days. Furthermore, higher serum hormone levels of ß-HCG, progesterone, and estradiol were reported in PE cases compared with normal cases. These increases were statistically significant in ß-HCG levels on Day-0, in progesterone levels on Day-0 and Day-1, and in estradiol levels on Day-1 and Day-3. Serum prolactin levels showed no significant differences at all. 

We also found that serum hormone levels of ß-HCG, progesterone, and estradiol were significantly reduced during study days in each study group separately. Systolic BP measurements in PE cases decreased significantly on Day-1 and Day-3 compared with Day-0 [Median (Range) (mm Hg): Day-0: 160 (130–200), Day-1: 157 (130–194), Day-3: 150 (130–190); *p* = 0.032 and *p* = 0.029 respectively)]. No significant differences in systolic BP measurements in the control group or in diastolic BP measurements in both groups were observed (Table 2).

Then, we calculated the percent ratio of observed differences in serum hormone levels as well as in BP measurements in order to scrutinize the scenario of these differences between study groups and within each group separately during study days. For this purpose, we included 20 control cases and 25 PE cases in which all measurements of all tested parameters were available. Our data showed that the biggest differences in serum hormone levels and in BP measurements between control cases and PE cases were reported on Day-0 (Figure 1A). In addition, we found that both study groups presented a similar schema of reduction percent in ß-HCG, progesterone, and estradiol serum levels during study days with maximal reduction percent between Day-0 and Day-3 (Figure 1B and Figure 1C). 

### 3.3. Association between Serum Hormone Levels with PE and BP Measurements

First, we tested the association between PE and parameters that differed significantly between the PE group and control group using logistic regression. We found that PE was negatively associated with gestation weeks and positively associated with creatinine concentrations, serum ß-HCG levels on Day-0, and serum progesterone levels on Day-1 (Table 3). However, only gestation age and creatinine concentrations showed as independent predictors of PE (data are not shown).

Next, as shown in Table 1, systolic and diastolic BP measurements were significantly higher in the PE group compared with the control group on all study days. Logistic regression was used to investigate various parameters that could predict these measurements. Serum ß-HCG levels on Day-0 were found to predict diastolic BP on Day-0 and systolic BP measurements on Day-3. Serum ß-HCG levels on Day-1 and Day-3 were found to predict systolic BP measurements on Day-3 (Table 4). When these observations were adjusted for other variables, they became no more statistically significant. We found that PE diagnosis was an independent predictor of systolic and diastolic BP measurements on all study days (data are not shown). 

### 3.4. Relevant Diagnosis and Characteristics within the PE Group

Figure 2 illustrates PE features among our patient women cohort. Briefly, severe PE and late-onset PE were reported in 54% and 57% of the patient women, respectively. Seven women were diagnosed with HELLP syndrome (23%) and sixteen women were characterised by IUGR (53%). 

Differences in serum hormone levels and BP measurements according to the pathological feature were tested using the Mann-Whitney test. Results showed that severe PE cases (N = 19) had significantly higher serum prolactin levels compared with mild PE cases (N = 11) on Day-0 [Median (Range) (µIU/mL): 7711 (2489–17,660) vs. 4702 (3056–9624), respectively, *p* = 0.030]. However, no significant differences in PB measurements according to disease severity were reported (Data are not shown). 

Next, we found that decreased progesterone and estradiol levels on Day-0 as well as increased systolic BP measurements on Day-3 were associated with worse PE features including early disease onset and IUGR diagnosis (Figure 3). Consequently, predictor variables for systolic PB on Day-3 in PE cases were tested using logistic regression. Data showed that IUGR diagnosis, early disease onset, and lower gestation age predicted increased systolic BP measurements on Day-3 [(β-*p*): (12.100–0.041), (−17.587–0.002) and (−1.559–0.021), respectively]. No significant results were reported for serum hormone levels (Data are not shown).

## 4. Discussion

PE has been extensively studied. Large cohort studies yielded a wealth of information on risk factors, symptoms, pathogenicity, clinical and laboratory findings, etc. PE characteristics such as obesity, nulligravida, and elevated liver enzymes and creatinine concentrations were confirmed in this study [28,29,30]. We also reported shorter gestation age in PE women. This is due to the fact that, to date, pregnancy termination is the only cure for this disorder [31]. 

PE, as previously stated, is classified as a hypertensive disease of pregnancy. Excessive maternal inflammatory response, caused by disrupted spiral artery remodeling in the placenta, provokes generalized maternal endothelial dysfunction, which contributes to the clinically visible elevations in maternal BP [32]. We found that despite a slight decrease in BP measurements during the puerperium, the significant increase in BP measurements in the PE group compared with the normal group persisted for the first three postpartum days. PE-induced hypertension is responsible for a substantial proportion of maternal mortality and for a wide range of postpartum complications that mandate long-term follow-up and contribute to the overall healthcare spending in developed countries [33]. Effective screening and intervention strategies for high BP in the postpartum period have been seriously considered [1,34,35]. To date, no specific biological marker has been identified that can accurately predict PE postpartum consequences. In a previous study, we assessed the predictive value of soluble fms-like tyrosine kinase 1 (sFlt-1), placental growth factor (PlGF), and their ratio to estimate the short-term postpartum maternal outcome. According to our findings, increased serum PlGF levels were associated with elevated systolic BP measurements. However, none of these markers could predict the general worsening of postpartum PE [26]. 

It is well known that hormonal changes, such as estradiol [36], progesterone [37], ß-HCG [38], and prolactin [39] are associated with the pathology of PE. In this study, we looked at whether the alteration schema in serum levels of the aforementioned hormones differs between the PE group and normal group over three postpartum days, and what value these biomarkers might have as predictors of PB measurements and poorer disease features. We found that, compared to control cases, PE cases had generally higher serum hormone levels. Only ß-HCG levels on Day-0 were significantly elevated, as were progesterone levels on Day-0 and Day-1, and estradiol levels on Day-1 and Day-3. These elevations were no more significant when adjusted to gestation age and creatinine concentrations. In addition, all serum prolactin level comparison tests were insignificant. As a result, with the current study design, it was not possible to determine the role of the tested hormones in PE pathogenicity. Nevertheless, this was not our study goal. For this purpose, further prospective studies in a large multicentric cohort are warranted. 

Next, we found that the alteration schema in serum hormone levels of estradiol, progesterone, and ß-HCG was similar in both study groups, with a gradual decrease over the course of the study (Figure 1B and Figure 1C), which could imply that the early postpartum profile of tested hormones is unaffected by the presence of PE. 

Furthermore, our findings revealed that serum levels of tested hormones had no predictive value for PB measurements, with the exception of ß-HCG levels, which predicted systolic on Day-3. However, when adjusted for PE diagnosis and gestational age, this effect was no more significant. In a retrospective case-control study, Chen and colleagues showed that free early pregnancy β-hCG levels in PE groups were significantly lower than in the control group. This study included 680 hypertensive disorders of pregnancy (HDP)-free pregnant women and 222 HDP-affected pregnant women, 90 of whom had PE and 71 of whom had severe PE. However, the risk calculation model developed using a combination of different factors including β-hCG after adjusting for maternal weight and gestation age outperformed the original method of using β-hCG alone [40].

Lan et al. found that, throughout pregnancy, the PE group had lower estradiol levels than the control group, but the two groups had similar progesterone levels [9]. Low levels of 2-methoxyestradiol, an estradiol metabolite, were found to be negatively correlated with systolic peak arterial pressure in patients with early-onset PE. On the contrary, in patients with late-onset PE, a rise in 2-methoxyestradiol serum levels has been reported. It has been proposed that this increase is due to the activation of compensatory mechanisms to keep normal serum estrogen levels. Exogenous administration of estradiol has been shown in animal models as well as in PE patients to normalize blood pressure and other associated symptoms of PE [41].

Prolactin has been linked to the pathogenesis of pregnancy-associated hypertension by modulating the activity of endothelial nitric oxide synthase [42]. Hypertensive pregnant women had higher blood prolactin levels than non-hypertensive pregnant women, but the difference was not statistically significant. Nonetheless, cord blood prolactin levels were significantly higher in babies born to hypertensive mothers than in normal pregnancy mothers [43].

As evidenced by the studies listed below, investigating better predictors of HDP has been a research hot spot in the field of prenatal medicine. As a result, there is no comparable data for our postpartum study. In this study, we assume that postpartum serum levels of the tested hormones could not be implicated in the pathogenesis of PE-induced hypertension.

Our data showed that serum prolactin levels in the PE group on Day-0 were lower by 40% in mild PE cases compared with severe PE cases. A reduction of 20% was also reported by Leaños-Miranda et al. who also found that urinary prolactin levels, as well as the presence of antiangiogenic prolactin fragments in urine, are associated with PE severity at the time of clinical manifestation [44]. Regardless, we found no significant differences in PB measurements based on disease severity, as well as no correlation between prolactin levels and PB measurements, despite the fact that early-onset disease and IUGR diagnosis could have predicted systolic BP measurements in our PE group on Day-3, and that these two subgroups had significantly lower serum estradiol and progesterone levels. While a causal relationship between these two hormone levels and BP measurements could not be established, it is tempting to propose that dysregulated serum prolactin, estradiol, and progesterone levels may be involved in poorer PE features via mechanisms unrelated to hypertension.

The small sample size limits the interpretation of our results. Although we were unable to establish a link between postpartum maternal hormone status and BP measurements, a larger number of cases could significantly refute or affirm our results. Another limitation is the notable difference in gestation age between study groups, which could affect the levels of tested hormones. Serum progesterone and estradiol concentrations in normal pregnant women, for example, increase with advanced pregnancy and reach 100-fold or more at delivery than before pregnancy [45]. The “snapshot” nature of single-point hormone testing is another limitation because hormone production follows a circadian rhythm, making it difficult to determine whether serum levels represent a peak, a valley, or something in between. As a result, larger, well-defined prospective studies are needed to demonstrate the postpartum pattern and actions of these hormones, as well as their role in predicting PE-related arterial hypertension.

## Figures and Tables

**Figure 1 diagnostics-12-01700-f001:**
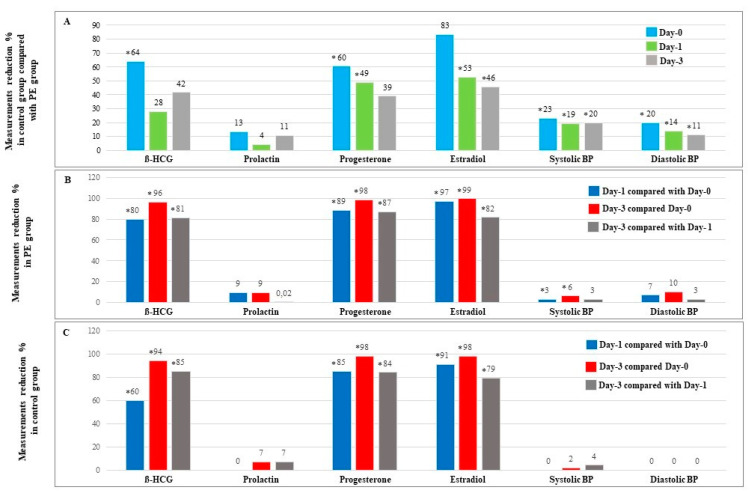
Measurements reduction % in serum hormone levels and BP measurements during study days. (**A**) Reduction of serum hormone levels and BP measurements in 20 normal cases compared with 25 PE cases. (**B**) Reduction of serum hormone levels and BP measurements in 25 PE cases. (**C**) Reduction of serum hormone levels and BP measurements in 20 normal cases with normal pregnancy. * Significant difference (*p* < 0.05 using Mann-Whitney-*U*-Test). Results are presented as percent.

**Figure 2 diagnostics-12-01700-f002:**
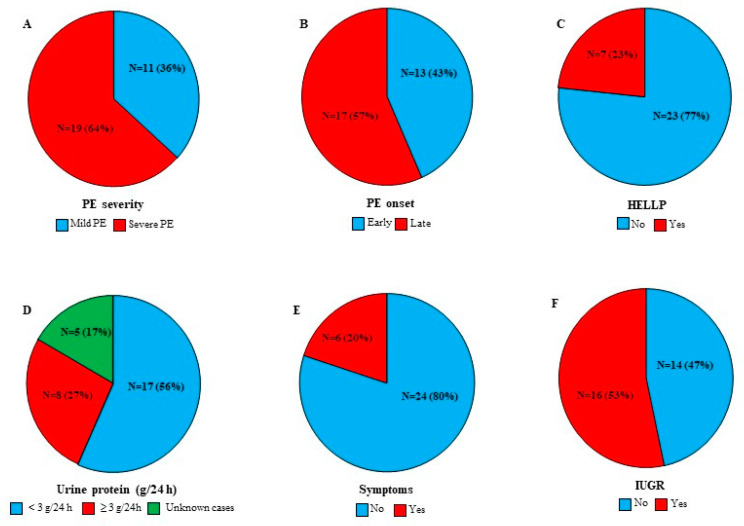
PE features among PE cases (N = 30). (**A**) PE severity. (**B**) PE onset. (**C**) HELLP cases. (**D**) Protein urea concentration. (**E**) Existence of one or more of the following signs: headache, severe edema, vaginal bleeding, epigastric pain, and vision changes. (**F**) IUGR cases. Results are presented as N (number of cases) (%).

**Figure 3 diagnostics-12-01700-f003:**
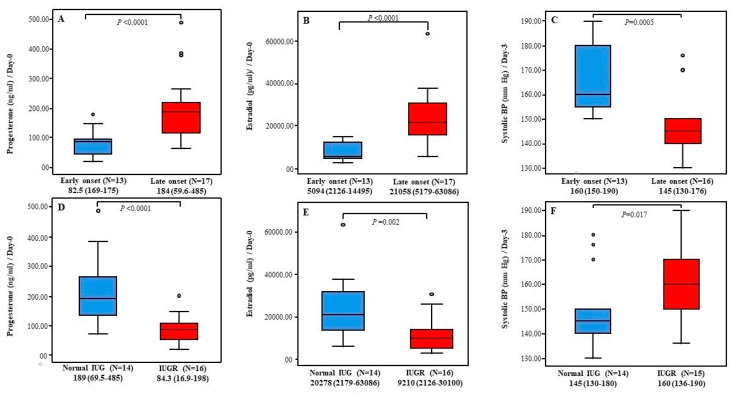
Postpartum serum progesterone and estradiol levels and systolic BP within PE cases. (**A**–**C**) Serum progesterone and estradiol levels at Day-0 and systolic PB measurements at Day-3 in cases with early PE onset vs. cases with late PE onset respectively. (**D**–**F**) Serum progesterone and estradiol levels at Day-0 and systolic PB measurements at Day-3 in cases with normal IUG vs. cases with IUGR respectively. Results are presented as Median (Range). *p*-value is tested using Mann-Whitney-*U*-Test. N: number of cases.

**Table 1 diagnostics-12-01700-t001:** Clinical and laboratory parameters of included cases.

Characteristic	Control Cases (N = 20)	PE Cases (N = 30)	*p*
* Age (Years)	31 ± 7	32 ± 5	NS
* BMI (Kg/m^2^)	32.3 ± 5.7	36.9 ± 8.7	0.030
⁑ Gestation age at the time of delivery (Weak)	38.7 (34.3–41.9)	34.4 (26.6–40.1)	<0.0001
* Creatinin (mg/dL)	0.6 ± 0.12	0.7 ± 0.16	<0.0001
⁑ Alanine aminotransferase (ALT) (U/L)	11.50 (8–28) ^φ^	20.5 (8–392)	0.003
⁑ Aspartate aminotransferase (AST) (U/L)	20 (15–34) ^φ^	31.5 (7–242)	0.018
^⁋^ Gravida [N (%)]			
One Gravida ≥2 Gravida	3 (15) 17 (75)	31 (70) 9 (30)	<0.0001
^⁋^ Para [N (%)]			
0 Para ≥1 Para	5 (25) 15 (75)	25 (83) 6 (17)	<0.0001
^⁋,^^‽^ Symptoms [N (%)]			
No Yes	20 (100) 0 (0)	24 (80) 6 (20)	NS

*: Results are shown as [Mean ± STD] and *p*-value is calculated using Student’s *t*-test. ⁑: Results are shown as [Median (Range)] and *p*-value is calculated using Mann-Whitney-*U-* Test. ^⁋^: Results are shown as number (%) and *p*-value is calculated using Chi-squared Test. ^‽^: Existence of one or more of the following signs: headache, severe edema, vaginal bleeding, epigastric pain, and vision changes. ^φ^: Data were available from 14 control cases. N: Number of cases. NS: Not significant.

**Table 2 diagnostics-12-01700-t002:** Serum hormone levels of ß-HCG, prolactin, progesterone, and estradiol and systolic- and diastolic BP measurements in control and PE cases on all study days.

Parameter	Control Cases (N = 20)	PE Cases ℷ	*p*
**Systolic blood pressure (mm Hg)**			
Day-0	123 (105–170)	160 (130–200)	<0.0001
Day-1	125 (100–145)	157 (130–194)	<0.0001
Day-3	120 (80–150)	150 (130–190)	<0.0001
*Φ p*	NS *	Day-1 vs. Day-0: 0.032 Day-3 vs. Day-0: 0.029 Day-3 vs. Day-1: NS	
**Diastolic blood pressure (mm Hg)**			
Day-0	80 (60–90)	100 (60–140)	<0.0001
Day-1	80 (50–90)	92 (70–120)	<0.0001
Day-3	80 (65–90)	90 (70–110)	<0.0001
*Φ p*	NS *	NS *	
**ß-HCG (mIU/mL)**			
Day-0	8357 (1786–95,328)	26,623 (2675–172,281)	0.005
Day-1	3379 (948–17,942)	4685 (487–43,860)	NS
Day-3	519 (6–2103)	916 (138–6975)	NS
*Φ p*	<0.0001 *	<0.0001 *	
**Prolactin (µIU/mL)**			
Day-0	5682 (1868–9277)	6327 (2489–17,660)	NS
Day-1	5687 (369–14,576)	5766 (314–10,661)	NS
Day-3	5310 (193–11,383)	5570 (149–12,275)	NS
*Φ p*	NS	NS	
**Progesterone (ng/mL)**			
Day-0	55.8 (8.16–250)	115.5 (16.9–485)	0.046
Day-1	8.3 (3.1–24.3)	14.8 (3.8–103)	0.002
Day-3	1.3 (0.2–3.7)	1.8 (0.8–11.7)	NS
*Φ p*	<0.0001 *	<0.0001 *	
**Estradiol (pg/mL)**			
Day-0	2544 (215–55,080)	12,675 (2126–63,086)	NS
Day-1	227 (75–817)	425 (85–10,665)	0.020
Day-3	48 (12–278)	89 (12–681)	0.008
*Φ p*	<0.0001 *	<0.0001 *	

ℷ: N = 30 at delivery day and Day-1; N = 29 at Day-3. *Φ*: *p*-value within each group. *: Significant differences between values from 3 days. N: Number of cases. NS: Not significant. Results are shown as [Median (Range)] and *p*-value is calculated using Mann-Whitney-*U*-Test.

**Table 3 diagnostics-12-01700-t003:** Association between PE and parameters that differ significantly between PE group and control group.

Predictor Variable *	B	*p*	Exp(B)	95% CI for Exp(B)	
				Lower	Upper
BMI ^Δ^	0.085	0.053	1.089	0.999	1.188
Gestation age (Week)	-0.324	0.004	0.723	0.581	0.900
Creatinin (mg/dL)	8.895	0.002	7294.2	25.7	20,741,143.8
ALT (U/L)	0.117	0.038	1.124	1.006	1.256
AST (U/L)	0.067	0.081	1.069	0.992	1.153
ß-HCG-Day-0 (mIU/mL)	0.031	0.030	1.03	1.003	1.061
Progesterone-Day-0 (ng/mL)	0.007	0.074	1.007	0.999	1.014
Progesterone-Day-1 (ng/mL)	0.145	0.018	1.156	1.025	1.304
Estradiol-Day-1 (pg/mL)	0.246	0.061	1.278	0.989	1.635
Estradiol-Day-3 (pg/mL)	0.100	0.109	1.100	0.978	1.250

*: Predictor variables included only parameters that were significantly different between PE group and control group. ^Δ^: Actual values.

**Table 4 diagnostics-12-01700-t004:** Simple linear regression of BP measurements with different parameters in all study cases.

	Systolic BP Measurements (mm Hg)			Diastolic BP Measurements (mm Hg)		
Predictor Variable	Day-0	Day-1	Day-3	Day-0	Day-1	Day-3
PE diagnosis						
*P*	<0.0001	<0.0001	<0.0001	<0.0001	<0.0001	<0.0001
β	37.800	29.7	36.4	21.8	13.4	13.6
BMI						
*P*	0.017	NS	NS	0.017	NS	NS
β	1.071	---	---	0.710	---	---
Gestation age (Weeks)						
*P*	<0.0001	<0.0001	<0.0001	0.005	0.024	0.002
β	−2.471	−2.467	−2.841	−1.475	−0.966	−1.060
Creatinin (mg/dL)						
*P*	0.001	0.003	0.003	0.003	0.018	NS
β	66.9	53.2	61.2	41.6	27.3	17.1
ALT (U/L)						
*P*	0.023	NS	0.040	NS	NS	NS
β	0.134	---	0.122	---	---	---
AST (U/L)						
*P*	NS	NS	0.042	NS	NS	NS
β	---	---	0.192	---	---	---
ß-HCG (mIU/mL) *						
Day-0 *P*	NS	NS	0.013	0.028	NS	NS
β	---	---	4.928	0.136	---	---
Day-1 *P*	NT	NS	0.004	NT	NS	NS
β	---	---	4.846	---	---	---
Day-3 *P*	NT	NT	0.014	NT	NT	NS
β	---	---	4.769	---	---	---

*: Only results of β-HCG were shown because the results of all other hormones were not significant. NS: Not significant. NT: Not tested.

## Data Availability

The data presented in this study are available on request from the corresponding author. The data are not publicly available due to privacy restrictions.

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
