# Peer review of "Postpartum Assessment of the Correlation between Serum Hormone Levels of Estradiol, Progesterone, Prolactin and ß-HCG and Blood Pressure Measurements in Pre-Eclampsia Patients"

_diagnostics, 2022, doi:10.3390/diagnostics12071700_

Round 1

Reviewer 1 Report

good study characteristic. an important study question. a study with a larger number of cases should be connected

Reviewer 2 Report

It is not clear why the hormonal levels were measured after delivery and not during the third trimester.

Why was prolactin and not testosterone measured?

The PE group is ok, but the control group is too small given the physiological variaiblity of the hormones. Why? It must be much more difficult to collect samples from 30 PE patients than from healthy pregnant women.

Within a comparison of two groups it should be "higher/lower" and not "increased/decreased" unless the authors mean dynamic changes between time points.

the effect size is not reported - by how much higher (%, fold) were the concentrations - this should be reported already in the abstract

I do not understand how the fact that the authors have found differences between healthy controls and PE patients might lead to the conclusion that the pathology is not of hormonal origin. The conclusion has to be rephrased.

The text should be revised "failer, begaining"

The diagnostic criteria should be clearly explained in this manuscript. A citation to a previous manuscript is not enough since this is a crucial part of the study. 

"medical records of patients"

Technical variability of the hormonal assays should be reported.

BMI should be reported as actual? prepartal? 

while it is clear why the gestation age differs between the groups, this could also be a reason for hormonal differences. This should be at least discussed. 

ALT and AST are named differently

Blood pressure should be reported first, then hormones.

Graphs should be prepared instead of tables. However, black and white graphs should show the variability and the dependence of before - after data points.

In addition to a comparison of groups the dynamic analysis should be conducted - 2-way RM-ANOVA is needed.

the multivariate analysis is important, but only if the number of subjects allows. in this case, this is not really informative. If the authors want to keep it, rather put it into the supplement.

abnormal distributions require medians and IQR, but also non-parametric tests.

Fig. 2 is not really informative.

I doubt that further clinical classification of patients is helpful in this study. The issue is again the relatively small sample size and, thus, rather small power of the study.

are there animal models that would support the role of the hormones in the pathogenesis?

The used literture should be updated since there are numerous relevant papers published in the last years which are not taken into account.
